# A Model Test for the Influence of Lateral Pressure on Vertical Bearing Characteristics in Pile Jacking Process Based on Optical Sensors

**DOI:** 10.3390/s20061733

**Published:** 2020-03-20

**Authors:** Yonghong Wang, Xueying Liu, Songkui Sang, Mingyi Zhang, Peng Wang

**Affiliations:** 1College of Civil Engineering, Qingdao University of Technology, Qingdao 266033, China; lxy0211ROSE@163.com (X.L.); 18306426194@163.com (S.S.); zhangmingyi@qut.edu.cn (M.Z.); pengw951@163.com (P.W.); 2Collaborative Innovation Center of Engineering Construction and Safety in Shandong Blue Economic Zone, Qingdao University of Technology, Qingdao 266033, China

**Keywords:** lateral pressure, axial force, model test, fiber grating, micro silicon piezoresistive sensor

## Abstract

Photoelectric integrated testing technology was used to study precast piles during pile jacking at the pile–soil interface considering the influence of the earth and pore water pressures on its vertical bearing performance. The low temperature sensitive fiber Bragg grating (FBG) strain sensors and miniature silicon piezoresistive sensors were implanted in the model pile to test the changes of earth pressure, pore water pressure and pile axial force of the jacked pile at the pile–soil interface, and the influence of lateral pressure on pile axial force was studied. The test results showed that the nylon rod is feasible as a model pile. The FBG strain sensor had a stable performance and monitored changes in the axial force of the model pile in real time. The miniature earth and pore water pressure sensors were small enough to avoid size effects and accurately measured changes in the earth and pore water pressures during the pile jacking process. During pile jacking, the lateral earth pressure increased gradually in depth, and the lateral earth pressure at the same depth tended to decrease at greater depths. Lateral pressures caused the axial force of the pile to increases by a factor of 1–2, where the maximum was 2.7. Therefore, the influence of the lateral pressure must be considered when studying the residual pile stress.

## 1. Introduction

In recent years, the rapid development of city construction has led to an increased density of high-rise buildings. As a result, precast piles have become the primary pile foundation for high-rise buildings. The application of precast piles is more extensive, while pipe piles that use the jacking method have low noise, high bearing capacity, strong pile body, and environmental protection and have been more widely utilized in recent engineering construction [1]. However, several aspects of jacked pipe pile research have been limited due to the difficult testing conditions and early stage research level. For example, the difficulty of embedding stress test elements in pipe piles has increased the difficulty of stress tests relative to cast-in-place piles.

The traditional method to test the internal force of a pile body is to use steel bar stress gauges. Togliani [2] and Igoe et al. [3] researched the load transfer mechanism and bearing performance of prefabricated piles with prestressed steel bars embedded in prefabricated piles and discussed the associated testing problems. With the rapid development of optical fiber testing technologies in recent years, Lee et al. [4] used the Brillouin method of optical fiber sensing to realize a distributed fiber strain test for prefabricated piles. Pando et al. [5] performed experimental research on the process to embed of a fiber grating sensing system in a jacking pile, which solved the problem of implanting a fiber FBG sensor in the pipe pile. Oh et al. [6] implanted a quasi-distributed optical fiber sensor in the pipe pile grooves to separate the side friction resistance and pile end resistance of the jacked pile penetration process to study the timeliness of the bearing capacity. Klar et al. [7] compared the results of the single pile static load test using either Brillouin distributed fiber technology or ordinary sensors and expounded the economics of the two approaches.

The traditional method to measure the earth and pore water pressures during the jacking process and the rest period after jacking is to embed earth and pore water pressure gauges in the affected soil layer at certain distances around the pile. Roy et al. [8] tested the pore water pressure for steady-state penetration. Pestana et al. [9] embedded earth and pore water pressure gauges in the soil around the pile to explore pressure changes in the soil at the pile end and certain distances around the pile during pile jacking. McCabe et al. [10] buried earth and pore water pressure sensors in the soil around the pile and presented a case history describing the measurements collected during the installation and load testing of groups of five, closely-spaced, precast, concrete piles in a soft clay-silt. Tang et al. [11] discussed the magnitude, distribution, and influence range of the lateral pressure increment and excess pore water pressure generated in the soil around a single pile during pile jacking by analyzing data measured during pile foundation construction. Li et al. [12] studied the variation mechanism of lateral earth pressure and deformation caused by excavation of existing composite foundation through centrifugal test.

The earth and pore water pressures at the pile–soil interface act directly on the pile body, which is more influential than when further from the pile. However, this kind of test was rarely reported in the literature [13], which limits the in-depth study of the pile–soil interface characteristics and bearing capacity mechanism. Li [14] measured the earth pressure coefficient on the pile side of the short pile using indirect methods and found that when the jacked pile was pressed, its pressure approached the passive earth value. This was between the static and passive earth pressures within the static load test, which is not the static earth pressure generally considered. The measurement method is to place two strain gauges at the same point of the pipe pile: one measures the vertical strain and the other measures the circumferential strain. The inner and outer pressures are calculated in reverse based on the Lame elastic mechanics of the ring subjected to both internal and external pressures. The Lame formula shows that the outer diameter pressure of the circular pipe affects the vertical axial force, which indicates that the radial earth pressure will produce an axial force on the pipe pile set in the foundation under a partial constraint.

As the earth and pore water pressures at the pile–soil interface act directly on the pile body, there are few reports on direct tests of earth and pore water pressures at the interface. This article utilized indoor tests with a model pile body surface groove encapsulation with micro silicon piezoresistive-type earth pressure sensors, pore water pressure sensors, and low-temperature sensitive, miniature, FBG, strain sensors. The earth pressure, pore water pressure, and pile body stress were tested during pile jacking close to the pile at the pile–soil interface. The effects of the earth and pore water pressures on changes to the pile and pile body stress were considered to analyze their influence on the lateral pile pressure on the vertical bearing performance. This provides a reference for subsequent research on the influence of radial pressure on the axial and residual stresses on the pile body.

## 2. Test Plan

### 2.1. Model Pile

Nylon rods have a high toughness, strong abrasion and shock resistances, strong tensile and bending strengths, minimal water absorptivity, good dimensional stability, etc., and can be easily processed at low costs. In this test, a nylon rod with a diameter of 60 mm and length of 750 mm was used as the model pile. The effective length of the model pile was 700 mm with a scale of approximately 1:10. A machine tool was used to carve grooves on the surface of the nylon rod (groove width × groove depth = 5 × 20 mm to install the hole manometer, 10 × 10 mm to install the soil manometer, and 3 × 3 mm to install the fiber grating string), as shown in Figure 1. The solid nylon rod was used as the model pile to simulate the closed-end pipe pile without considering the effects of the earth plug.

### 2.2. Model Bucket and Loading Device

The model bucket has a diameter of 800 mm and a height of 1200 mm and is welded from a 1.5 mm thick steel plate. The diameter of the model bucket is approximately 13.3 times the diameter of the pile, so the influence of boundary effect can be ignored [15,16,17]. The clamping guide device was welded to the model bucket to ensure the vertical support of the pile during the penetration process. Three scale sticks were pasted into the model bucket at angles of 120° to facilitate the production of the model foundation in later stages.

In the test, the reaction frame and hydraulic jack were used as the test loading devices, where the reaction frame was assembled in the structure laboratory. The four corners were fixed to the ground using high-strength bolts, the jack was fixed on the beam of the reaction frame with a steel strand, and the model barrel was placed under the reaction frame to ensure the centrospheres of the model barrel and jack coincided. The model bucket and loading device are shown in Figure 2.

### 2.3. Sensor and Package

#### 2.3.1. Basic Principle of Sensor

Fiber Bragg grating (FBG) is a type of optical sensor that combines germanium, tin, boron and other photosensitive elements into the core during optical fiber fabrication to change the optical sensitivity of the optical fiber. The phase grating is written into the grating by means of ultraviolet irradiation to make the refractive index of the fiber core change periodically. When the broadband light source is injected into the fiber, the light satisfying the diffraction condition of the Bragg grating is reflected, and the diffraction condition is Equation (1).
(1)$$\lambda_{B}=2n_{\mathrm{eff}}\Lambda$$
where, $\lambda_{B}$ is the central wavelength of reflected light, $n_{\mathrm{eff}}$ is the refractive index of grating, and Λ is the grating spacing.

Silicon piezoresistive type soil pressure sensor is based on the high sensitivity coefficient of the piezoresistive effect of polycrystalline silicon material, silicon diaphragm pressure make into four insulation silica piezoresistive sensor, by wheatstone bridge silicon diaphragm four varistor different voltage output, according to the output voltage of silicon diaphragm strain value, to determine the sensitivity coefficient of sensor.

#### 2.3.2. Installation of Sensors

The experiment adopted a low-temperature sensitive, miniature, FBG, strain sensor testing model for the pile strain (or stress), and micro silicon piezoresistive-type earth and pore water pressure sensors for the model pile to determine the earth and pore water pressures. The FBG strain sensor could automatically eliminate the influence of temperature changes, which negates the need for compensation. Images of the three sensors are shown in Figure 3.

Three low-temperature sensitive, miniature, FBG, strain sensors sized at Φ1 × 20 mm were connected with armored fiber in series. The sensors were placed in the reserved groove of the model pile and sealed with epoxy resin. The part of the fiber that was connected to the fiber grating demodulator was protected with a hollow steel sleeve to prevent damage caused by excessive bending and positions of the three sensors, as shown in Figure 4, where the sensors at the *a-a* section are close to the pile top.

The miniature earth and pore water pressure sensors are both characterized by direct contact between the elastic element and the measured medium and exhibit easy miniaturization, which is suitable for dynamic force measurements. The size of the five pore water pressure sensors was Φ8 × 15 mm, and the size of the five earth pressure sensors was Φ8 × 15 mm. The two types of pressure sensors had the same distribution positions on the model pile, as shown in Figure 5. This is because the sensor at the measuring head must be flush with the pile surface of the pile body for stress testing. The thin steel formwork was the initial sensor position, which was sealed with a plastic film to prevent clogging from the subsequent epoxy resin. The epoxy resin encapsulation level was adjusted before solidification for complete curing while the plastic film on the top of the test was removed. The model pile after encapsulation is shown in Figure 5. To test the pile end resistance during the model pile jacking, the micro silicon piezoresistive earth pressure sensor was encapsulated by drilling holes at the end of model pile, as shown in Figure 6.

### 2.4. Model Foundation

In actual engineering, most of the soil layers inserted by the pipe piles under a static pressure are viscous. Therefore, the silty clay of the construction site in Qingdao was selected to prepare the model foundation. Soil samples retrieved from the site were tested in the laboratory to determine its physical and mechanical parameters, as shown in Table 1.

The retrieved clay was dried and ground in an oven and crusher for later use. The moisture content of the model foundation was set to 28% based on the testing requirements. The production steps of the model foundation were as follows:(1)First, apply sealant to the weld in the model barrel to prevent water from overflowing from the weld.(2)Make the model foundation in 10 layers with a 10 cm thickness for each layer, and calculate the mass of dry soil and water required for each layer of the foundation based on the dry density of the soil, the volume of the model bucket, and the moisture content of the model foundation.(3)Evenly spread the weighed dry soil into the model bucket using the pasted scale, and use the indoor light tamper to pack the soil the design height with a smooth surface. Spray the weighed water evenly into the soil with the sprinkling kettle. After sprinkling, seal the mouth of the bucket with a plastic sheet and let it stand for 24 hours to allow the water to fully soak into the soil.(4)Repeat the above steps until the foundation of the model is complete. Seal the model barrel mouth with a plastic sheet for 7 days to ensure all the soil layers are uniform. The process to make the model foundation is shown in Figure 7.

### 2.5. Test Process

In practical engineering, the silty clay layer is usually not located at the surface of the foundation. To better model actual pile driving processes and to avoid the shallow piling force being too small, 40 pieces of 5 kg weights were uniformly pressed on the model foundation before the start of the trial. A plastic film was placed between the weights and model foundation pad, which was equivalent to approximately 400 kPa of overload to simulate the upper soil.

(1)Before the test, make marks every 5 cm on the model pile, and number the miniature earth and pore water pressure sensors from the bottom to the top to record the test data.(2)Bring the equipment required for the test to the structural laboratory, paste the pressure sensor to the center of the end face of the hydraulic jack using structural adhesive, and connect the data display.(3)Connect the low-temperature sensitive, miniature, FBG, strain sensor to the fiber Bragg demodulator through the fiber extension cord, which is connected to the computer through the data cable. Connect the connector of the miniature earth and pore water pressure sensors to the data acquisition card and external switching power supply, which is connected to another computer.(4)Connect the hydraulic jack to the oil pump with the oil pipe, place the model pile at the center of the model foundation, and adjust the position of the jack so that its center and the center of the model pile are on the same line.(5)Adjust the time of the two computers in advance and set the relevant parameters of the testing software to ensure synchronized data collection from the two sensors. Start pressurizing with the oil pump and turn on the data acquisition software to start recording data. Record the pile force from the data display every 5 cm of pressing. When the jack is full, add a pad to continue driving the pile until the end of the test. The entire test device is shown in Figure 8.

## 3. Test Results and Analysis

As the tests were indoors and the piling time was short, the effects of temperature causing a wavelength change in the FBG strain sensor was ignored. What the FBG strain sensor can obtain is the wavelength change of the sensor. The pile stress can be calculated using Equations (2) and (3), and the axial force of the pile can be obtained by multiplying the sectional area of the pile with the stress.
(2)$$\sigma =E\times \frac{\Delta \lambda \times 10^{-3}}{K_{\varepsilon }}$$
where, $K_{\varepsilon }\approx 1.2pm/\mu \varepsilon$, $K_{\varepsilon }=\left\{1-\frac{n^{2}}{2}\left[p_{12}-\upsilon (p_{11}+p_{12})\right]\right\}\lambda$, *p*_11_ and *p*_12_ are photoelastic effect constants. *v* is poisson’s ratio; *ε* is the axial strain of the fiber. *E* is the elastic modulus (GPa) of the model pile, Δ*λ* is the wavelength change (nm), and *K* is the strain sensitivity coefficient of the miniature, FBG, strain sensor (pm/*με*).
*N_ij_* = *σ_ij_* · *A_ij_*(3)
where, *A* is the cross-sectional area of pile (cm^2^).

The experiment manually loaded the model pile penetration in the grave and maintained a uniform piling process. However, the penetration rate was still dynamic and was difficult to maintain at a fixed value. The penetration rate during pile jacking was approximately 2–3 cm/min. According to Bond et al. [18], the penetration rate of the classification method is considered to be rapid.

### 3.1. Variation Law of Pile Pressure with Penetration Depth

During the entire testing process, the pressure sensor pasted on the hydraulic jack was displayed through the data display instrument in real time. The pressure curve of the pile with the penetration depth during the penetration process is shown in Figure 9.

As seen from Figure 9, the penetration resistance at the initial stage of the pile pressing was relatively small. Changes in the pile pressing force were small within penetration depths of 14–36 cm. As greater penetration depths, the pile pressing force continued to increase. When the pile was pressed vertically into the soil, the soil around the pile underwent severe squeezing disturbances, and a pore water pressure was generated in the soil. The soil within a certain range around the pile produced a remodeling zone, which significantly reduced the shear strength of the soil. As the pile jacking progressed, the excess pore water pressure in the soil began to slowly dissipate, and the shear strength gradually recovered, which increased the penetration resistance. On the other hand, as the pile jacking progressed, the penetration depth of the pile body became increasingly larger, and the side friction resistance and penetration resistance of the jacking pile both increased. Due to the limited travel of the jack, after the model pile penetrated a certain depth, it was necessary to add a pad to achieve continuous penetration. After the pad was added, the starting force was greater than the pile pressing force before the pad. Excess pore water pressure was generated when the soil around the pile was disturbed, and a water film was generated between the pile and the soil [19]. The model pile was in a standing state when adding cushions, and the water film reduced slightly when the pile was standing. Therefore, the pile jacking resistance increased when the pile settled after standing, which is similar to the actual pile connection during pile jacking.

### 3.2. Distribution of Pore Water Pressure

Changes in the pore water pressure as measured by each miniature, silicon, piezoresistive, pore water pressure sensor when the model pile was pressed through different depths is shown in Figure 10. The pile disturbs the soil during the static pressure pile penetration process. The soil within a certain range around the pile produces a remodeling zone where a pore water pressure is generated. During the test, changes in the hydrostatic pressure were small and ignored. Changes in the pore water pressure were a result of excess pore water pressure. As seen from Figure 10, at the beginning of the model pile penetration, the increasing rate of the excess pore water pressure was relatively slow. However, at deeper penetration depths, the pore water pressure increased approximately linearly. When the penetration depth was 35 cm (half of effective pile length), the pore water pressure reached a maximum value before slowly dissipating. Under different penetration depths, the excess pore water pressure at the same soil layer differed, which is explained as follows. Increased penetration depths leads to a slightly lower excess pore water pressure at the same soil layer, but the change was not significant. The development law for excess pore water pressure is consistent with the research results of Tang et al. [11], Zhu [20], and Dash et al. [21] on excess pore water pressure in the soil around piles. Hwang et al. [22] believed that no matter it is a silt layer (6 m) or a clay layer (9 m), the pore water pressure at the same horizontal depth during the jacking process changes almost simultaneously. However, the excess pore water pressure immediately before the pile–soil interface began to dissipate earlier than in previous works.

### 3.3. Lateral Pressure Distribution

Similarly, as the penetration depth of the model pile increased, changes in the earth pressure as measured by each miniature, silicon, piezoresistive, earth pressure sensor is shown in Figure 11. It can be seen from Figure 11 that the lateral pressure of the pile in the cohesive soil during pile jacking penetration gradually increased with the penetration depth. In the initial stage of pile jacking, the lateral pressure grew relatively fast. As the pile jacking progressed, the lateral pressure growth rate slowed down. At greater pile depths into the soil, the lateral pressure at the same depth changed and showed a decreasing trend. That is, the lateral pressure of the pile was the largest when it was just pressed into the cohesive soil. At greater depths of the model pile, the lateral pressure gradually decreased, but the reduction was larger. When the model pile was just pressed into the cohesive soil layer, the soil mass was strongly compressed, resulting in a large lateral pressure. However, at greater penetration depths of the pile, the pore water pressure gradually decreased, the effective earth pressure gradually increased, shear deformation occurred, and the soil particles were rearranged, which gradually decreased the side pressure.

The excess pore water pressure was small during the entire test, accounting for 1–2% of the lateral pressure with a maximum of 3%. Combining Figure 10 and Figure 11 provides the relationship between the effective lateral pressure and penetration depth, which is shown in Figure 12. The figure indicates that the trend of the effective side pressure curve was nearly consistent with the change trend of the side pressure curve. However, the value was slightly reduced due to the pore water pressure.

### 3.4. Distribution of Lateral Friction Resistance

The pile jacking resistance was obtained from the pressure sensor attached to the jack. The miniature earth pressure sensor installed at the end of the model pile measured the resistance at the end of the pile, and the distribution of the lateral friction resistance of the model pile was obtained through calculations, as shown in Figure 13.

As seen from Figure 13, the lateral friction increased with the penetration depth. The lateral friction increased rapidly at the initial stage of pile jacking but increased more slowly at greater penetration depths. The frictional resistance was nearly unchanged at depths from 24 to 40 cm. Considering changes in the pore water pressure with depth, the excess pore water pressure had the fastest growth rate and reached a maximum value within the considered range. The pile model caused a relatively significant soil disturbance, which greatly reduced the extrusion of soil shear strength. At the same time, the model generated a water film between the pile and soil, which reduced the side friction. At greater pressing depths, the excess pore water pressure slowly dissipated, while the lateral friction began to increase. Within a pressing depth range of 64–70 cm, the lateral friction remained nearly unchanged.

### 3.5. Axial Force Distribution Law for Pile Body

The axial force distribution on the pile body can be calculated from the pile pressure and the calculated lateral friction resistance. This distribution can also be measured with the low-temperature sensitive, miniature, FBG, strain sensor embedded in the pile body along with Equation (1), as shown in Figure 14.

It is seen from Figure 14 that the axial force of the three sections increased at greater pile depths, but the axial force distribution obtained from the measured pile strain using the FBG strain sensor was different from that obtained from the calculations. As seen from the axial force curve measured with the FBG sensor, there was an obvious strain change at the *a-a* section at the initial pile compression, and the axial force continued to increase with the pile compression depth. There was an initial strain change at the *b-b* section in the grating demodulation instrument. However, at a pile penetration of about 5 cm (1/14 of the effective length) a slowly growing wavelength change appeared until approximately 24 cm (1/3 of the effective length). After this depth, the axial force increased significantly with the depth. Excess pore water pressure was generated in the soil due to the extruding disturbance from the pile during the compaction process, which resulted in a decreased shear strength in the soil and a decrease in the frictional resistance to the pile. In this stage, more pile pressure was converted into an axial force from the pile body, resulting in a significant increase in the axial force at the *b-b* section. The axial force at the *c-c* section was similar to the *b-b* section. A strain change did not occur until the pile penetration was approximately 9 cm (1/8 of the effective pile length). With an increased penetration depth, the axial force continued to grow, but at a relatively low rate. The calculated distribution curve for the axial force was nearly consistent with the measured distribution law. The axial forces at the *b-b* and *c-c* sections changed at the initial stage of pile jacking, which was different from the results of the sensor tests. The small pile pressure at the initial stage of pile jacking and the large elastic modulus of the nylon rod (average elastic modulus of 2.6 GPa) caused part of the compression deformation of the model pile itself to be lost during the load transfer process. This resulted in constant strain when the load was transferred to the middle and lower parts of the pile.

Studies have shown that lateral pressures produce both circumferential and vertical stresses on the pile [12], and the measured axial force under a lateral pressure was greater than the calculated value. However, it is also seen from Figure 14 that the measured axial force obtained at the initial stage of pile compression was less than the calculated axial force. According to the depth and width of groove in the pile, the volume of epoxy resin in the groove was calculated to be 156,750 mm^2^, and the ratio of epoxy resin to nylon rod in the pile was calculated to be about 7:93, so as to determine the weighted average of the elastic modulus of epoxy resin and nylon rod. The axial stress of the pile was calculated using Equation (1) considering that the epoxy resin occupies a certain proportion in the model pile. The elastic modulus obtained by the above calculation method is substituted into Equation (1) for calculation. The loss model for pile deformations and sensor errors led to the early pile axial force being less than the calculated value. 

The differences between the measured and calculated values at the three sections were inconsistent. The differences at the *b-b* and *c-c* sections were small, while the difference at the *a-a* section was relatively large. In the early stages of pile jacking, only the middle and lower parts of the pile were affected by the lateral earth pressure. After the model pile was inserted 35–40 cm, the measured axial force was greater than the calculated value. Differences between the measured and calculated values also varied at different soil depths. The measured axial force was larger, with an amplitude 1–2 times the effective lateral pressure at the same depth, where the maximum was a factor of 2.7. Since the calculated value in this paper was the axial force in the traditional sense, this increase was caused by the lateral earth pressure.

During the pile jacking process, the force locked inside the pile body after the completion of the one-way pile pressing is called the residual stress due to the unloading of the pile top and the failure to fully recover the elastic compression of pile body, which has been comprehensively considered [23,24,25,26,27,28]. The test shows that for partially constrained pile bodies, the lateral pressure affects the vertical stress, and there was a considerable increment of the vertical stress as caused by the pile body. Therefore, the influence of the lateral pressure should be considered when studying the residual stress of pile bodies. That is, a portion of the residual stress is caused by the lateral pressure, which has not been addressed in previous studies. Similarly, this effect should also be considered when measuring and calculating the pile axial force.

## 4. Conclusions

This paper studied the earth pressure, pore water pressure, and axial force of a pile body at the pile–soil interface in a viscous soil using a laboratory model test. Preliminary conclusions and suggestions are as follows:(1)A low-temperature sensitive miniature FBG strain sensor can monitor stress changes of a pile in real time, which is easily installed and has a stable performance. The miniature earth and pore water pressure sensors are small and suitable for dynamic measurements. For the first time, a micro FBG strain sensor, micro earth pressure sensor, and micro pore water pressure sensor were installed in the grooves of a model pile (nylon rod) to mechanically test the pile and verify its feasibility.(2)In this test, silty clay retrieved from the site was successfully dried and ground to make the model foundation; thus, the soil sample preparation method is feasible.(3)In the clay soil, the change law for the interfacial force at the pile–soil interface is as follows. During the pile jacking process, the earth pressure on the pile side increased gradually with depth and began to change rapidly. As the depth increased, the earth pressure gradually slowed. The lateral pressure at the same depth tended to decrease at greater pile depths. The distribution law for the effective lateral pressure was nearly consistent with that of the lateral pressure. Under the test, the excess pore water pressure during pile jacking was small and accounted for about 1%–3% of the total lateral pressure. Moreover, the pore water pressure was maximized when the pile penetration depth was half the effective pile length.(4)Under the influence of the lateral pressure, the measured pile axial force increased compared with the traditional calculated value at approximately 1–2 times the effective lateral pressure at the same depth, where the maximum was a factor of 2.7. Therefore, the influence of lateral pressure should be considered when studying problems such as the pile residual stress.(5)The foundation soil of this experiment is homogeneous viscous soil. The variation rules of lateral pressure and vertical axial force in other soil layers and the relationship between them need to be further studied.

## Figures and Tables

**Figure 1 sensors-20-01733-f001:**
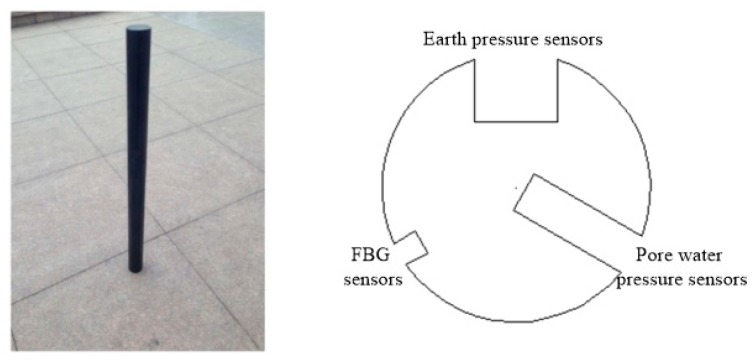
The model pile.

**Figure 2 sensors-20-01733-f002:**
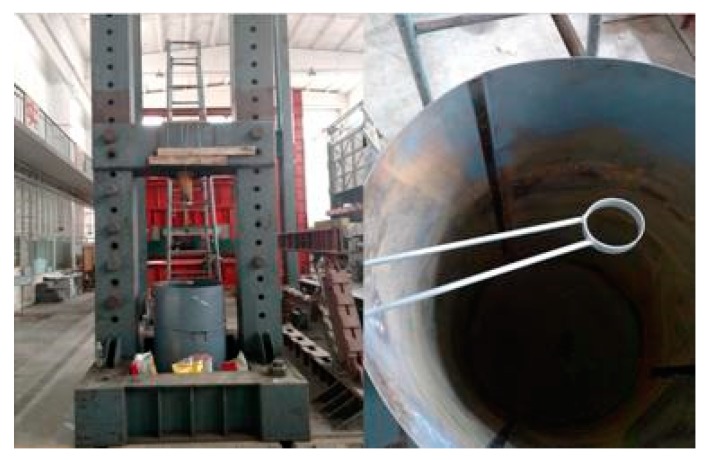
Model bucket and loading unit.

**Figure 3 sensors-20-01733-f003:**
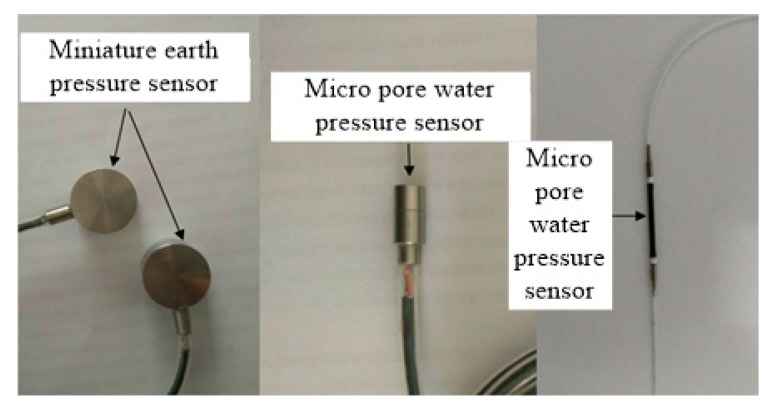
Photographs of the three sensors used in the experiments.

**Figure 4 sensors-20-01733-f004:**
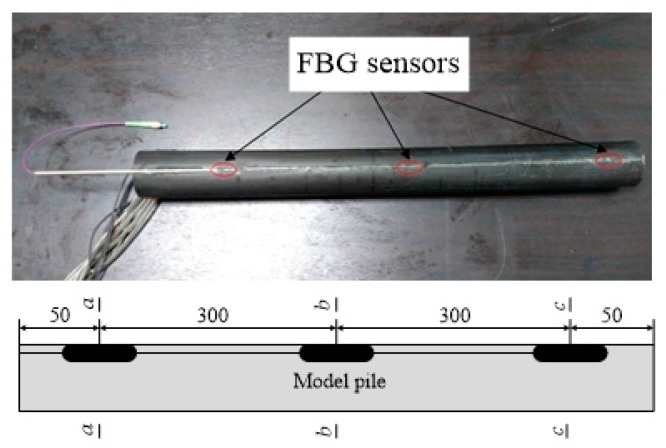
Photograph after encapsulating the FBG sensors and position of the FBG sensors in the model.

**Figure 5 sensors-20-01733-f005:**
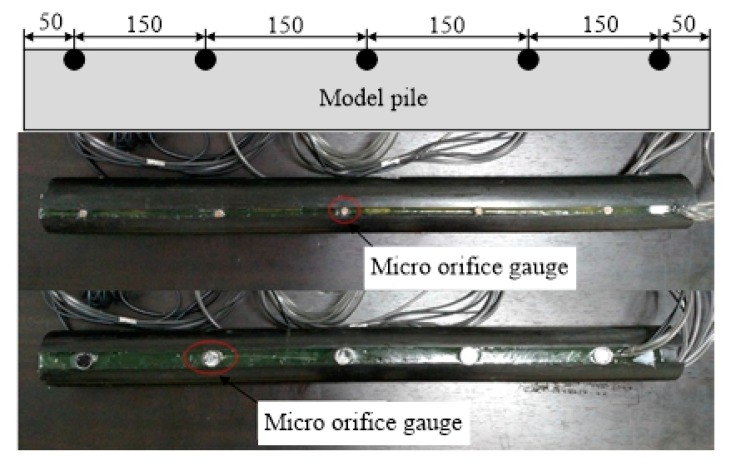
Positions of the micro earth and pore water pressure sensors and photographs after the encapsulated of the micro earth and pore water pressure sensors.

**Figure 6 sensors-20-01733-f006:**
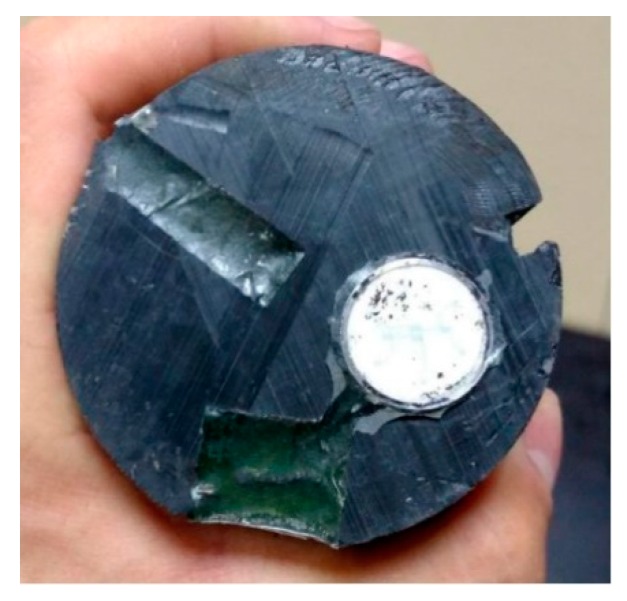
Micro earth pressure sensor in the pile end.

**Figure 7 sensors-20-01733-f007:**
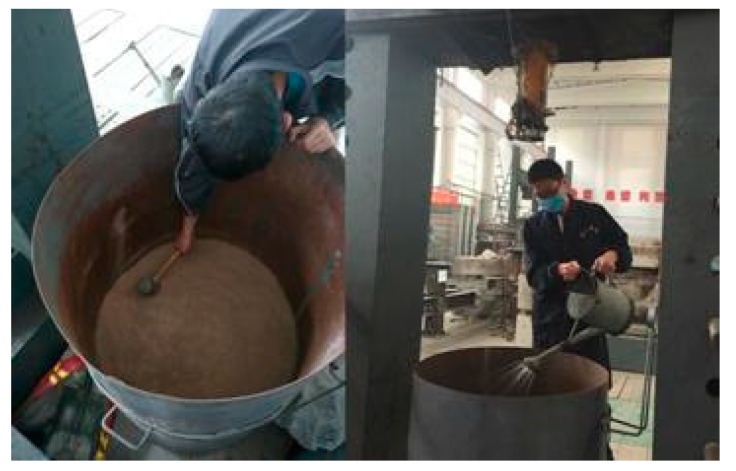
Working process for the model foundation.

**Figure 8 sensors-20-01733-f008:**
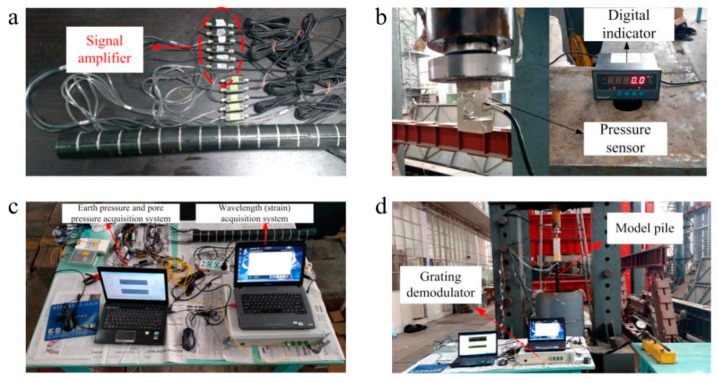
Photographs showing the testing process: (**a**) Marked model pile and sensors. (**b**) Pressure sensors and matching digital display. (**c**) Connected instruments and equipment. (**d**) Testing and data collection.

**Figure 9 sensors-20-01733-f009:**
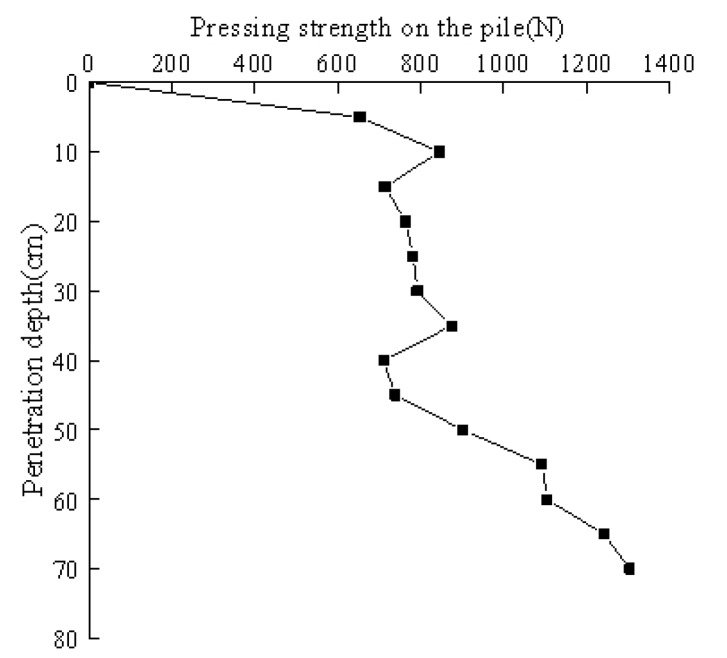
The relationship between the pile driving pressure and the penetration depth at different pile pressing depths.

**Figure 10 sensors-20-01733-f010:**
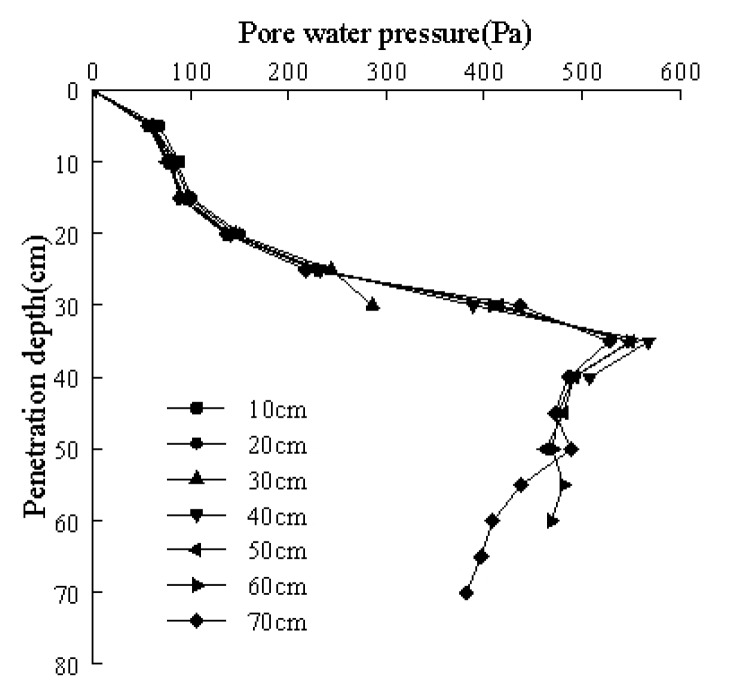
Relationship between the pore water pressure and penetration depth at different pile pressing depths.

**Figure 11 sensors-20-01733-f011:**
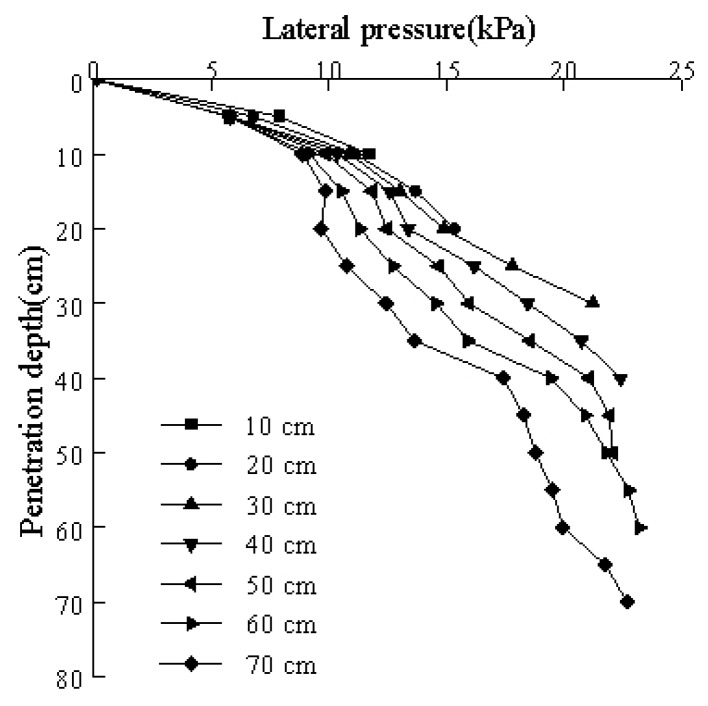
Relationship between the lateral pressure and penetration depth at different pile pressing depths.

**Figure 12 sensors-20-01733-f012:**
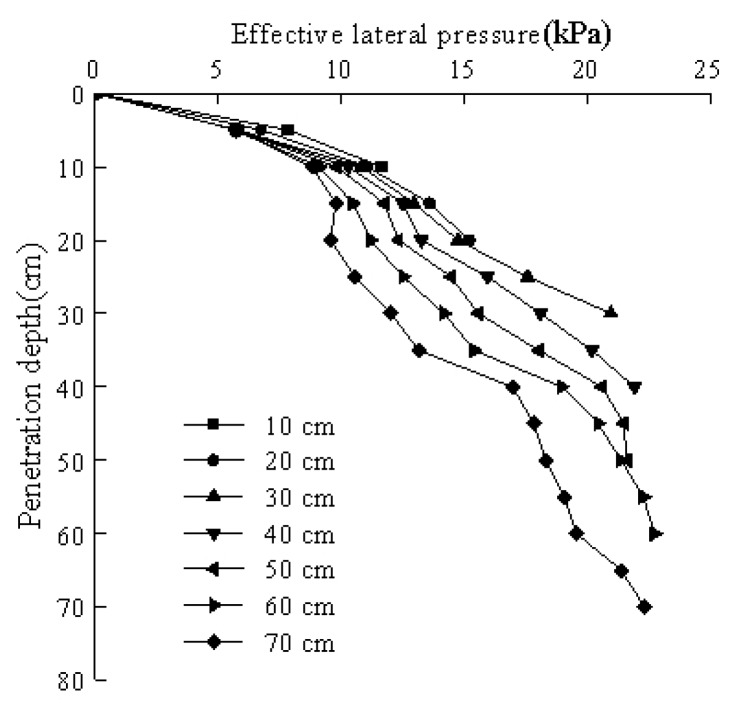
The relationship between the effective lateral pressure and the penetration depth at different pile pressing depths.

**Figure 13 sensors-20-01733-f013:**
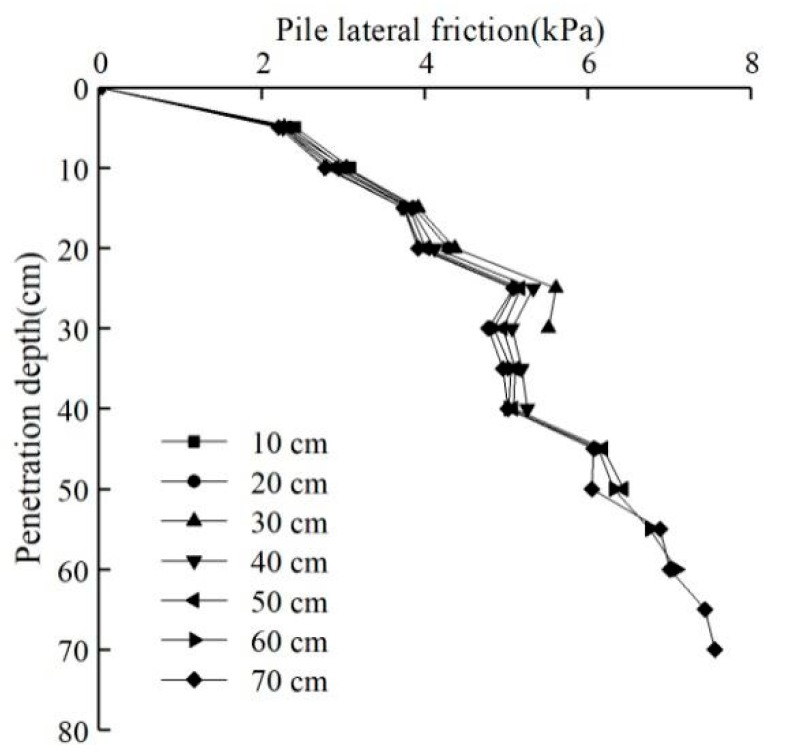
The relationship between the lateral friction and penetration depth at different pile pressing depths.

**Figure 14 sensors-20-01733-f014:**
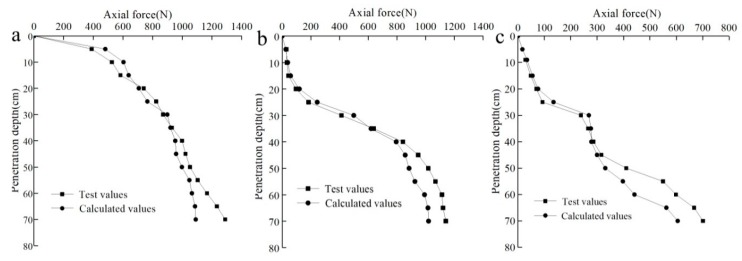
The relationships between the axial force and the penetration depth: (**a**) *a-a* section, (**b**) *b-b* section, (**c**) *c-c* section.

**Table 1 sensors-20-01733-t001:** Physical and mechanical parameters of the tested soil.

Density (g/cm^3^)	Moisture Content (%)	Dry Density (g/cm^3^)	Void Ratio	Saturability (%)	Liquid Limit (%)	Plastic Limit (%)	internal Friction Angle (°)	Cohesion (kPa)	Coefficient of Compressibility *α*_v1-2_ (MPa^−1^)	Modulus of Compression *E*_S1-2_ (MPa)	Poisson’s Ratio
1.98	25.3	1.58	0.728	94.9	31.3	16.5	30	27	0.32	5.5	0.3

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
