# Peer review of "A Model Test for the Influence of Lateral Pressure on Vertical Bearing Characteristics in Pile Jacking Process Based on Optical Sensors"

_sensors, 2020, doi:10.3390/s20061733_

Round 1
Reviewer 1 Report
The authors present a applications, in Civil Engineering, usin optical sensors based in FBG. In recentes years several proposes were presented. The authors present the possibility of na optical sensor using FBG to measure the influence of lateral pressure on vertical bearing characteristics during pile jacking. The manuscript is interesting but need some clarification and change. Following some suggestion to improve the manuscript.
- Change the title. The use of optical sensors needs in the title.
- The abstract need to be rewrite. Is confuse. The authors present some results but is confuse.
- In introdution, one final paragraph where the authors explain the work.
- In the model the authors needs explain clearly each parameters. Why don’t use different diameter of model pile? This is important.
- The authors not explain clearly the influence of soil in results. Expalin the use of micro water sensor
- The authors present the model and the results for different parameters. But the correspondence to the real case is missing. The pilar model used corresponds to a real pilar with what dimensions? Explain the influence in the results because the materials are diferents : concrete and iron in real situation versus Nylon rods.
- What is the influence of temperature in the results.
- One comparation with standard sensors and this configuration is necessary. In figure 3 the authors show but not refers more in the results. In figure 6 show in the end of pilar but the sensor needs in the same position of FBG.
- In the manuscript, the authors need clarify each kind of test and present the results clearly.
- The different figures needs a improve in the presentation.
- The conclusion needs to be improve.
I recommend major review of this manuscript.
Reviewer 2 Report
The paper is related to experimental analyses of pile jacking process. The subject is interesting, but some shortcomings have to be explained prior to publication.
Remarks:
1) Fig. 1. The figure have to be corrected as in the present version it is hard to see the differences between the model pile stages. It will be valuable to indicate in the photograph changes in the element geometry. The figure caption is confusing.
2) The measurements were performed using low-temperature sensitive FBG strain sensor. How the temperature sensitivity of FBG sensor was reduced? What is a gage length of the sensor?
3) Page 7. The sentence: “The FBG strain sensor measured the wavelength change of the sensor.” is unclear.
4) Eq. 1. What is a value of K coefficient and what is it relationship to particular Bragg wavelength? The sentence: “delta lambda is the wavelength change of the model pile penetration process (nm)” is unclear in relation to FBG sensors working principle.
5) Are the results presented in sections 3.1.-3.4. just a raw measurements form sensors or they are an effect of a signal processing?
6) Fig. 14. Are the test results just measured values from the testing machine? How the relationship between axial force and penetration depth for FBG sensors was calculated using eq. 1, as it is a relationship between stress and wavelength change?
7) Page 11. It will be better to describe process in relation with strain instead of wavelength change. Strain is a common parameter in mechanical engineering, while wavelength change is related to sensor working principle.
8) Page 11. “The epoxy resin and nylon rods were both incorporated into the formula to calculate the weighted average of the elastic modulus, rather than simply using the model pile material elastic modulus.” What weights were used during calculations?
9) Page 11. “The axial stress of the pile was calculated using formula (1) considering the cross section of the encapsulated sensor at a certain proportion of the epoxy resin.” Firstly, axial stress is not present. Secondly, what does it mean “certain proportion”?
The paper is not recommended for publication in the presented form.
Round 2
Reviewer 1 Report
The authors presente in the coverleter the response to the question in the previous report. However, before publish is necessary the author introduce in the manuscript the effect of temperature and change the graphics
Reviewer 2 Report
Thank you very much for the explanations. Although, there are still some shortcomings that have to be corrected prior to publication.
Remarks:
1) Fig. 1. It still does not provide any important information. It will be better to introduce locations of the geometrical changes of the pile.
2) What is a principle of operation of the sensor. How the temperature influence is limited? It should be explained.
3) Eq. 1. The explanation related to K epsilon should be introduced into the text.
4) Eq. 2. What is calculated in Eq. 2?
5) Be careful using ‘x’ symbol in equations as it is a symbol related to a particular type of mathematical operation - vector product.
6) Are all sensors presented in the paper based on FBG technology? If they are, the principle of operation of every sensor should be presented.
7) Information about the weights using during the elastic modulus calculation should be introduce into the paper text.
8) There is still no explanation what does it mean ‘certain proportion’. Such information should be provided into the paper text.
Round 3
Reviewer 2 Report
Thank you very much for the explanations. The paper was improved and now can be accepted for publication.